# First Synthesis of *(−)*-Altenuene-D_3_ Suitable as Internal Standard for Isotope Dilution Mass Spectrometry

**DOI:** 10.3390/molecules24244563

**Published:** 2019-12-12

**Authors:** Michael A. Sebald, Julian Gebauer, Thomas Sommerfeld, Matthias Koch

**Affiliations:** 1HPC Standards GmbH, Am Wieseneck 7, D-04451 Borsdorf, Germany; 2AnalytiCon Discovery GmbH, Hermannswerder Haus 17, D-14473 Potsdam, Germany; j.gebauer@ac-discovery.com; 3Bundesanstalt für Materialforschung und –prüfung, Abteilung Analytische Chemie, Referenzmaterialien, Richard-Willstätter-Strasse 11, D-12489 Berlin-Adlershof, Germany; thomas.sommerfeld@bam.de (T.S.); matthias.koch@bam.de (M.K.)

**Keywords:** altenuene, *Alternaria* mycotoxins, food safety, isotope-labeled, SIDA-LC-MS/MS, Suzuki coupling

## Abstract

Metabolites from *Alternaria* fungi exhibit a variety of biological properties such as phytotoxic, cytotoxic, or antimicrobial activity. Optimization of a literature procedure culminated in an efficient total synthesis of *(−)*-altenuene as well as a stable isotope-labeled derivative suitable for implementation in a LC-MS/MS method for mycotoxin analysis.

## 1. Introduction

### 1.1. Alternaria Mycotoxins

Invading crops at the pre- and post-harvest stage, certain filamentous fungi can contaminate food and feedstuff by producing a variety of toxic secondary metabolites, which are referred to as mycotoxins [1]. Derived from the extremely wide-spread genus, *Alternaria*, the commonly named *Alternaria* toxins [2,3] are frequently found in agricultural crops, such as grains, fruits, and vegetables, as well as soil, wall papers, and textures, and have thus been implicated in several animal and human health disorders. The European Food Safety Authority (EFSA) assessed the risks for public health related to the presence of *Alternaria* toxins in food and feed. Although the toxicological data on various *Alternaria* toxins was limited, it recommended the supervision of those toxins in foods and feeds [4]. Considering the need for possible future regulation, a suitable LC-MS/MS standard method for the determination of the most relevant *Alternaria* metabolites is required (Figure 1).

With *(−)*-altenuene (ALT, **1a**) [5] being the most acutely toxic in mice (LD50 > 50 mg/kg) [4], the development of new analytical methods for its detection and quantification has become of great importance for human and animal health risk assessment. Due to its high selectivity, sensitivity, and multi-analyte suitability, LC-MS/MS has become the method of choice for trace analysis of mid polar and polar organic contaminants in food and feed. As a major step forward in the improvement of accuracy (trueness and precision), isotope labeled compounds (mostly ^13^C-, ^2^D- or ^15^N-labeling) are widely used as internal standards to quantify the target analytes by LC-MS/MS. This so-called stable isotope dilution assay (SIDA, SIDA-LC-MS/MS) is considered the primary ratio method, representing a high level of metrology. Thus, the development of isotope labeled standards for mycotoxins and their implementation in LC-MS/MS methods have received much attention over recent years [6].

Moreover, **1a** has recently been reported to exhibit interesting cytotoxic activity against HCT116 cell lines with an IC50 value of 3.13 µM. This makes it a potential lead compound for the development of new anti-tumor drug candidates [7], which further supported our interest in working towards an efficient and reliable access to ALT (**1a**) and ALT-D_3_ (**1b**) by total synthesis.

### 1.2. Retrosynthetic Analysis

Based on the only total synthesis of **1a** reported to date [8], we herein wish to present the first preparation of a deuterated *(−)-*altenuene derivative (ALT-D_3_, **1b**) following an improved procedure. To prevent any ‘cross talk’ between the native and the labeled analyte we aimed to synthesize ALT-D_3_ (**1b**) by coupling the deuterated boronate **4b** and halogenated allylic alcohol **5b**, which should permit a facile isotope incorporation using commercially available reagents in an analogous fashion as described for the native compound **1a** (Scheme 1) [8].

## 2. Results and Discussion

### 2.1. Synthesis of Deuterated Boronate 4b

The synthesis of the deuterated boronic acid derivative **4b** was efficiently achieved in analogy to a literature procedure [9] but starting rather with a regioselective alkylation of the 4-hydroxy group using commercial iodomethane-D_3_ (99.5atom% D) as the deuterium source (Scheme 2). The slightly lower yield compared to the Mitsunobu reaction employed by Podlech et al. is compensated by the simplicity and the low cost of the reagents used in this protocol.

### 2.2. Synthesis of Bromo Alcohol 5b

A major drawback in the original synthesis of **1b** is the unfavorable diastereoselectivity of ~1:6 in the Grignard reaction affording the required tertiary alcohol **5a** (Scheme 1) [8]. In order to tackle this issue, we first decided to optimize the key carbonyl addition reaction. With the labile iodo substrate **10a** limiting the number of applicable reagents, we envisioned the usage of the much more stable bromo enone derivative **10b**, which should allow us to investigate other organometallic reagents. Thus, bromination of the known enone **9** obtained in 4 steps from inexpensive D-*(−)*-quinic acid (**8**) according to the literature, gave bromo enone **10b** with an excellent overall yield (Scheme 3).

We then focused on the optimization of the inefficient methylation of halo enones **10a** and **10b** (Table 1). Applying the original reaction conditions (Table 1, Entry 1) to the bromo enone **10b** provided a ~1:4 mixture of **5b** and the undesired isomer **epi-5b** (Table 1, Entry 2). A further refinement to a ~1:2 ratio could be achieved by the addition of stoichiometrical amounts of CeCl_3_ (Table 1, Entry 3). Interestingly, changing from the methyl Grignard reagent to the more reactive methyllithium, the diastereoselectivity changed completely with the desired carbinol **5b** now being the major isomer (*d.r.* ~ 1.4:1, Table 1, Entry 6). While the addition of CeCl_3_ proved to be disadvantageous in this case (Table 1, Entry 7), lowering the reaction temperature to –78 °C resulted in the best diastereomeric ratio obtained with ~1.7:1 in favor of **5b** (Table 1, Entry 8) but came at the cost of incomplete conversion of the starting material **10b**. Applying the same conditions to iodo enone **10a** only nonspecific decomposition was observed as expected (Table 1, Entry 9), emphasizing the initially proposed enhanced chemical stability of the bromo derivative **10b**.

Any attempts to further improve the diastereoselectivity of the 1,2-addition by employing either chiral catalytic systems (e.g., BINAP + Josiphos, which is actually known to promote 1,4-addition [11]) and/or alternative methyl donating reagents (e.g., AlMe_3_, DABAL, ZnMe_2_) [12,13,14] resulted in the formation of either the 1,4-addition product or only traces of the desired alcohol **5b** (Table 1, Entries 4,5,10–14).

On a preparative more useful scale, the newly established conditions (Table 1, Entry 8) delivered the requisite bromo alcohol **5b** with an essentially improved yield of 38% after chromatographic separation of both isomers by MPLC. With substantial amounts of the labeled boronate **4b** and the crucial alcohol **5b** in hand we started assembling the pieces.

### 2.3. Suzuki Coupling of Bromo Alcohol 5b and Boronate 4b

Alcohol **5b** and boronate **4b** were subsequently subjected to the reported Suzuki cross coupling conditions (Pd(OAc)_2_, S-Phos, Cs_2_CO_3_, dioxane/H_2_O, 80 °C) [8], approved for the arylation of iodo alcohol **5a**. Unfortunately, this test almost exclusively resulted in homodimerization of the boronic ester **4b**. After thoroughly scouring the literature for alternative catalytic systems, which would permit the desired hetero coupling to proceed, we were delighted to find that a simple alteration of catalyst and base (Pd(dppf)Cl_2_, NEt_3_, THF/H_2_O, 70 °C) [15] facilitated a smooth conversion, providing the advanced intermediate **11b** with 81% isolated yield (Scheme 4).

Interestingly, only traces of concomitant lactonization product **12b** were observed illustrating the remarkable mildness of this adjusted protocol. Cyclization using K_2_CO_3_ in methanol and cleavage of the bisketal protecting group by refluxing in aqueous AcOH finally gave rise to **1b** in 81% yield after two steps and purification by preparative HPLC (purity > 99.9% and > 99atom% D). Lastly, submitting alcohol **5b** and the unlabeled boronate **4a** (synthesized analogously using iodomethane [9]) to the same reaction sequence delivered ALT (**1a**) with identical yields.

### 2.4. Implementation of the ALT-D_3_ Standard (1b) in a LC-MS/MS Method

There are some recent LC-MS/MS methods available for the quantification of *Alternaria* toxins including ALT (**1a**) in food and feed based on positive or negative ionization mode as well using acidic or alkaline LC conditions [16,17,18]. Due to better performance data, ESI(−) mode under alkaline LC conditions was used to set up/optimize the MS/MS and LC parameters of the synthesized ALT-D_3_ (**1b**) (Table 2). HPLC: Agilent 1200 with autosampler; column: Eurospher 100-5 C18 P, particle size 5 µm, 250 × 4 mm (Knauer, Berlin, Germany); Inj.-vol: 10 µL; oven temp.: 30 °C; flow: 0.5 mL/min; eluent A: water with 5 mM ammonium acetate and ammonium hydroxide (pH 8.7), eluent B: Methanol with 5 mM ammonium acetate. 0–5 min 90% A, 5–22 min 0% A, 22–30 min 90% A. MS/MS: AB-Sciex QTrap4000; turbo ion spray, single reaction monitoring (SRM; negative polarity); TEM: 500 °C, CUR: 50 a.u., CAD: 12 a.u., IS: −2000 V; DP: −60 V; CE: −40 V; CXP: −10 V.

ALT-D_3_ (**1b**) does not show any signals for the mass transition of native ALT (**1a**) and vice versa with identical retention times (t_R_ (**1a/1b**) = 16.67 min, Figure 2). This is important for ideal compensation of ionization effects (mostly matrix suppression effects) and the use of ALT-D_3_ (**1b**) as internal standard. Moreover, the presented SIDA-LC-MS/MS method does not only allow the analysis of ALT (**1a**) and ALT-D_3_ (**1b**) but is also applicable to other relevant *Alternaria* toxins, e.g., alternariol (**2a**), alternariol monomethyl ether (**2b**), tentoxin or tenuazonic acid (**3**).

## 3. Materials and Methods

Commercial chemicals and solvents were used as received without any further purification. Triethylamine was dried over KOH, distilled in vacuo, and stored under an atmosphere of nitrogen. All reactions were carried out under an inert gas atmosphere using dry grade reagents and solvents unless stated otherwise. Reactions were monitored by thin-layer chromatography on Merck TLC Silica gel 60 F_254_ sheets with UV-visualization (254 nm and 336 nm) or KMnO_4_ staining. The diastereomeric ratios of the compounds **5** and **epi-5** were determined using a hp Series II 5860 GC device (SGE Analytical Science column, 25 m × 0.22 µm, BP × 5 × 0.25 µm, HP Inc., Palo Alto, CA, USA; Trajan Scientific, Ridgewood, Victoria, Australia) connected to a hp 5971 Series mass selective detector. Melting points were determined with the MP 90 melting point device by Mettler Toledo (Columbus, OH, USA). MPLC purification was performed with a Shimadzu MPLC system (Shimadzu Corp., Kyōto, Japan). The conditions and devices used for LC-MS/MS analysis of compounds **1a** and **1b** are stated in Section 2.4. ESI-high resolution mass spectra were recorded with a Bruker Daltonik micrOTOF coupled with a LC Packings Ultimate HPLC system (Bruker Corp., Billerica, MA, USA; Dionex/LC Packings, Sunnyvale, CA, USA). NMR spectra were either recorded on a Varian Mecury Plus 300 (300.8 MHz), Varian Mercury Plus 400 (399.95 MHz), or a Bruker Avance III HD (400.13 MHz) spectrometer (Varian/Agilent Technologies Inc., Santa Clara, CA, USA; Bruker Corp., Billerica, MA, USA). All signals were referenced to the respective solvent signals reported in the literature [19]. All coupling constants *J* refer to hydrogen–hydrogen interactions unless stated otherwise. The ^1^H and ^13^C NMR spectra of all new compounds as well as UV/Vis- and IR-spectra of the native and labeled natural products can be found in the Appendix A.

### 3.1. Synthesis of (2S,3S,4aR,8aR)-7-bromo-2,3,4a,5-tetrahydro-2,3-dimethoxy-2,3-dimethylbenzo[b][1,4]dioxin-6(8aH)-one (**10b**)

Enone **9** (2.91 g, 12.0 mmol, 1.00 equiv) was dissolved in 32 mL DCM in a 250 mL round-bottom flask and cooled to 0 °C. A solution of 632 µL (1.96 g, 12.3 mmol, 1.02 equiv) bromine in 32 mL DCM was added slowly over 1 h with a dripping funnel and the mixture was stirred for another 30 min at the same temperature. After that, 2.85 mL (1.96 g, 20.4 mmol, 1.70 equiv) NEt_3_ was added and the resulting blue solution was warmed to room temperature while stirring. After 1 h GC-MS analysis indicated the complete consumption of the starting material and the reaction was quenched with NaHCO_3_ (100 mL) and the phases were separated. The aqueous phase was extracted with DCM (3 × 75 mL), and the combined organic phases were washed with brine (50 mL), dried over Na_2_SO_4_, and the solvent was evaporated. The crude product was purified by chromatography using a CH/EE-gradient (15 →25% EE) yielding 3.3 g (86%) of the title compound **10b** as a white solid. R_f_ (CH/EE, 4:1) = 0.38. MP = 202 °C (decomposition). HR-MS: Calc. for [M + Na]^+^ = 343.0152, found: 343.0161. ^1^H-NMR (300 MHz, CDCl_3_): δ = 1.32 (s, 3 H), 1.36 (s, 3 H), 2.59 (dd, *J* = 13.5, 16.4 Hz, 1 H), 2.94 (dd, *J* = 4.8 Hz, 16.4 Hz, 1 H), 3.26 (s, 3 H), 3.31 (s, 3 H), 4.07 (ddd, *J* = 4.8, 9.1, 13.5 Hz, 1 H), 4.48 (ddd, *J* = 0.3, 2.0, 9.1 Hz, 1 H), 7.30 (dd, *J* = 0.3, 2.0 Hz, 1 H). ^13^C-NMR (75 MHz, CDCl_3_): δ = 17.7, 17.8, 41.3, 48.4, 48.5, 67.6, 70.4, 100.0, 101.1, 124.4, 148.9, 188.8.

### 3.2. Synthesis of (2S,3S,4aR,6R,8aR)-7-Bromo-2,3,4a,5,6,8a-hexahydro-2,3-dimethoxy-2,3,6-trimethylbenzo[b][1,4]dioxin-6-ol (**5b**)

To a solution of bromo enone **10b** (1.0 g, 3.1 mmol, 1.0 equiv) in 62 mL dry THF (0.05 M) at −78°C was added methyllithium (1.6 M in Et_2_O, 3.0 mL, 4.8 mmol 1.5 equiv) and the reaction mixture was slowly warmed to r.t. over 1 h. Saturated NH_4_Cl-solution (40 mL) was added and the aqueous phase was extracted with MTB (3 × 50 mL). The combined organic phases were dried (Na_2_SO_4_) the solvent was evaporated. Purification of the crude product by normal-phase chromatography (YMC-Gel, 6 nm S-15 µm) using MTB/heptane (3:7) and evaporation of the solvents yielded 374 mg (38%) of alcohol **5b** as a light yellow solid. R_f_ (*n*-Hep/MTBE, 7:3) = 0.18. GC-MS: [M − OMe]^+^ = 305/307, t_R_ = 17.05 min. MP = 73.9 °C. HRMS (ESI): Calc. [M + Na]^+^ = 359.0465, found: 359.0475. [α${]}_{D}^{22}$(c = 0.1, CHCl_3_) = +9. ^1^H-NMR (400 MHz, CDCl_3_): δ = 1.30 (s, 3 H), 1.33 (s, 3 H), 1.42 (s, 3 H), 1.83 (t, *J* = 13.1 Hz, 1 H), 2.11 (s, 1 H), 2.22 (dd, *J* = 3.6, 13.3 Hz, 1 H), 3.26 (s, 3 H), 3.27 (s, 3 H), 3.93 (ddd, *J* = 3.6, 8.9, 12.7 Hz, 1 H), 4.10 (dd, *J* = 1.7, 8.9 Hz, 1 H), 6.04 (d, *J* = 1.8 Hz, 1 H). ^13^C-NMR (101 MHz, CDCl_3_): δ = 17.90, 17.94, 30.5, 40.1, 48.1, 48.2, 65.4, 70.6, 73.5, 100.2, 100.7, 130.5, 131.3.

### 3.3. Synthesis of 5-Hydroxy-7-methoxy-2,2-dimethyl-4H-benzo[d][1,3]dioxin-4-one-D_3_ (**7**)

1.01 g (4.81 mmol, 1.00 equiv) 5,7-dihydroxy-2,2-dimethyl-4H-benzo[d][1,3]dioxin-4-one **6** and 731 mg (5.29 mmol, 1.10 equiv) K_2_CO_3_ were suspended in 15 mL acetone (0.2 M) in a 100 mL screw-top flask. After stirring at room temperature for 10 min, the flask was charged with 330 µl (767 mg, 5.29 mmol, 1.10 equiv) CD_3_I and the mixture was refluxed for 3 h in the sealed vessel. The suspension was cooled to room temperature, acetone was removed in vacuo, and the residue suspended in 100 mL EtOAc. The suspension was washed with water (3 × 50 mL), and the aqueous phase was extracted with EtOAc (2 × 30 mL). The combined organic phases were washed with brine (1 × 50 mL), dried over sodium sulfate and the solvent was removed. Normal-phase chromatography with PE/EE 9:1 (*v/v*) delivered 804 mg (74%) of compound **7** as an off-white solid. R_f_ (CH/EE, 4:1) = 0.36. MP = 102.9 °C. ^1^H-NMR (400 MHz, CDCl_3_): δ = 1.73, (s, 6 H), 6.00 (d, *J* = 2.3 Hz, 1 H), 6.14 (d, *J* = 2.3 Hz, 1 H), 10.44 (s, 1 H). ^13^C-NMR (101 MHz, CDCl_3_): δ = 25.8, 93.2, 94.8, 95.9, 107.0, 157.0, 163.3, 165.3.

### 3.4. Synthesis of 7-Methoxy-2,2-dimethyl-4-oxo-4H-benzo[d][1,3]dioxin-5-yl trifluoromethanesulfonate-D_3_

476 µl (796 mg, 2.82 mmol, 1.50 equiv) Tf_2_O was added to an ice-cold solution of 428 mg (1.88 mmol, 1.00 equiv) of phenol **7** in 3.8 mL dry pyridine (0.5 M) dropwise over 5 min, and the mixture was stirred at 0 °C. After 1 h, the GC-MS analysis indicated the complete consumption of the starting material. The solvent was removed in vacuo, and the residue was dissolved in 100 mL EtOAc. The solution was washed with CuSO_4_ (4% in H_2_O, 2 × 50 mL), H_2_O (2 × 50 mL), and brine (1 × 50 mL). The combined organic phases were dried over Na_2_SO_4_, and the solvent was evaporated. Chromatography with cyclohexane/acetone 7:1 (*v/v*) yielded 628 mg (82%) of the title compound as a pale yellow solid upon removal of the solvents. R_f_ (CH/EE, 4:1) = 0.19. MP = 58.5 °C. HRMS (ESI): Calc. [M + Na]^+^ = 382.0258, found: 382.0282. ^1^H-NMR (400 MHz, CDCl_3_): δ = 1.74 (s, 6 H), 6.48 (d, *J* = 2.4 Hz, 1 H), 6.53 (d, *J* = 2.4 Hz, 1 H). ^13^C-NMR (101 MHz, CDCl_3_): δ = 25.7, 101.3, 105.5, 106.7, 117.3, 120.5, 150.1, 157.2, 159.0, 165.7.

### 3.5. Synthesis of 7-methoxy-2,2-dimethyl-5-(4,4,5,5-tetramethyl-1,3,2-dioxaborolan-2-yl)-4H-benzo[d][1,3]dioxin-4-one-D_3_ (**4b**)

7-methoxy-2,2-dimethyl-4-oxo-4H-benzo[d][1,3]dioxin-5-yl trifluoromethanesulfonate-D_3_ (404 mg, 1.12 mmol, 1.00 equiv) was dissolved in a solution of 470 µL (341 mg, 3.37 mmol, 3.00 equiv) freshly distilled NEt_3_ in 10 mL dry dioxane, and the mixture was degassed with N_2_ for 15 min. Then 65 mg (56 µmol, 5.0 mol%) Tetrakis(triphenylphosphin)palladium and 485 µL (432 mg, 3.37 mmol, 3.00 equiv) 4,4,5,5-Tetramethyl-1,3,2-dioxaborolane were added successively, and the mixture was stirred at 80 °C for 2 h. After cooling down, the solvent was evaporated, and the crude product was directly submitted to manual flash chromatography with toluene/acetone 19:1 (*v/v*) yielding 277 mg (73%) of the title compound **4** as an orange solid without any impurification by the reduced side product. R_f_ (CH/EE, 4:1) = 0.22. MP = 102.1 °C. HRMS (ESI): Calc. [M + Na]^+^ = 360.1668, found: 360.1681. ^1^H-NMR (400 MHz, CDCl_3_): δ = 1.42 (s, 12 H), 1.71 (s, 6 H), 6.38 (d, *J* = 2.3 Hz, 1 H), 6.66 (d, *J* = 2.4 Hz, 1 H). ^13^C-NMR (101 MHz, CDCl_3_): δ = 24.9 (4 C), 26.0 (2 C), 84.6, 101.7, 106.3, 108.8, 113.8, 157.8, 162.0, 165.7.

### 3.6. Synthesis of 5-((2S,3R,4aR,6R,8aR)-2,3,4a,5,6,8a-hexahydro-6-hydroxy-2,3-dimethoxy-2,3,6-trimethylbenzo[b][1,4]dioxin-7-yl)-7-methoxy-2,2-dimethyl-4H-benzo[d][1,3]dioxin-4-one-D_3_ (**11b**)

To a degassed solution of alcohol **5b** (250 mg, 0.74 mmol, 1.00 equiv), boronate **4b** (312 mg, 0.92 mmol, 1.50 equiv), freshly distilled NEt3 (2.19 mL, 1.59 g, 15.7 mmol, 21.2 equiv) in 8.2 mL (0.09 M) THF/H_2_O 9:1 (*v/v*), Pd(dppf)Cl_2_∙CH_2_Cl_2_ (48 mg, 60 μmol, 8.0 mol%) was added, and the reaction mixture was stirred for 2 h at 70 °C before the solvents were evaporated. The residue was redissolved in a small amount of DCM and directly purified by FC using EtOAc/cyclohexane 1:2 (*v/v*). Evaporation of the solvents furnished 279 mg (81%) of the deuterated coupling product **11b** as a white solid. R_f_ (CH/EE, 2:1) = 0.17. MP = 165–185 °C (decomposition). HRMS (ESI): Calc. [M + Na]^+^ = 490.2127, found: 490.2121. [α${]}_{D}^{22}$ (c = 0.1, MeOH) = +1.5. ^1^H-NMR (300 MHz, DMSO-D_6_): δ (ppm) = 1.20 (s, 6 H), 1.66 (s, 6 H), 1.69 (s, 1 H), 1.84 (m, 1 H), 3.15 (s, 3 H), 3.19 (s, 3 H), 3.95–4.09 (m, 2 H), 5.04 (s, 2 H), 5.20 (s, 1 H), 6.59 (d, *J* = 2.5 Hz, 1 H), 6.77–6.70 (m, 1 H). ^13^C-NMR (75 MHz, DMSO-D_6_): δ = 17.6, 17.8, 24.5, 25.8, 27.2, 42.3, 47.3, 47.4, 65.2, 69.7, 71.0, 99.1, 99.8, 100.8, 104.9, 105.6, 112.6, 112.6, 124.6, 142.1, 144.7, 157.9, 159.0, 162.9.

### 3.7. Synthesis of (6aR,7aR,9S,10S,11aR)-4-Hydroxy-2,9,10-trimethoxy-7a,9,10-trimethyl-6a,7,7a,9,10,11a-hexahydro-5H-benzo[c][1,4]dioxino[2,3-g]-chromen-5-one-D_3_ (**12b**)

To a solution of the coupling product **11b** (240 mg, 0.51 mmol) in MeOH (10 mL), K_2_CO_3_ (78 mg, 0.6 mmol, 1.1 equiv) was added, and the reaction mixture was stirred for 1 h at r.t. before the solvent was evaporated. Saturated NH_4_Cl solution (15 mL) was added, and the aqueous phase was extracted with EtOAc (3 × 20 mL). The combined organic phases were dried (Na_2_SO_4_), and the solvent was evaporated. Chromatography with n-hexane/ethyl acetate 8:1 (*v/v*) provided 217 mg (100%) of the protected ALT derivative **12b** as a white solid. R_f_ (*n*-Hex/EE, 8:1) = 0.05. MP = 158–160 °C. HRMS (ESI): Calc. [M + Na]^+^ = 432.1708, found: 432.1705. [α${]}_{D}^{22}$ (c = 0.1, CHCl_3_) = +6. ^1^H-NMR (400 MHz, CDCl_3_): δ = 1.33 (s, 3 H), 1.34 (s, 3 H), 1.48 (s, 3 H), 1.90 (dd, *J* = 12.9, 14.5 Hz, 1 H), 2.49 (dd, *J* = 4.5, 14.5 Hz, 1 H), 3.26 (s, 3 H), 3.32 (s, 3 H), 3.87 (ddd, *J* = 4.5, 8.9, 13.2 Hz, 1 H), 4.25 (dd, *J* = 1.7, 8.9 Hz, 1 H), 6.14 (d, *J* = 1.7, 1 H), 6.42 (d, *J* = 2.4 Hz, 1 H), 6.49 (d, *J* = 2.4 Hz, 1 H). ^13^C-NMR (101 MHz, CDCl_3_): δ = 17.90, 17.9, 28.4, 38.9, 48.28, 48.33, 66.2, 69.6, 81.5, 100.0, 100.5, 100.6, 100.9, 103.0, 128.8, 133.9, 139.0, 164.2, 166.3, 169.0.

### 3.8. Synthesis of (2R,3R,4aR)-2,3,4,4a-Tetrahydro-2,3,7-trihydroxy-9-methoxy-4a-methylbenzo[c]chromen-6-one-D_3_ ((−)-altenuene-D_3_, **1b**)

The bisketal protected natural product **12b** (150 mg, 0.37 mmol) was heated to 100 °C while stirring in 5.2 mL of a 4:1 mixture of AcOH and H_2_O for 2 h after which the solvents were evaporated. Residual acetic acid was removed by successive addition and evaporation of DCM (two or three times). Then the crude product was purified by RP chromatography on a C_18_ stationary phase with H_2_O/MeOH 4:6, delivering 87 mg (81%) of the deuterium labeled natural product **1b** as a white solid. R_f_ (DCM/MeOH, 20:1) = 0.18. MP = 117–120 °C. LC-MS (neg): [M – H]^−^ = 294, t_R_ = 16.67 min. HRMS (ESI): Calc. [M + Na]^+^ = 318.1027, found: 318.1031. [α${]}_{D}^{22}$(c = 0.03, MeOH) = −9. ^1^H-NMR (400 MHz, DMSO-D_6_): δ = 1.47 (s, 3 H), 1.95 (dd, *J* = 7.3, 14.0 Hz, 1 H), 2.26 (dd, *J* = 3.5, 14.0 Hz, 1 H), 3.64–3.76 (m, 1 H), 3.91–4.00 (m, 1 H), 5.13 (d, *J* = 3.8 Hz, 1 H), 5.29 (d, *J* = 6.2 Hz, 1 H), 6.30 (d, *J* = 3.3 Hz, 1 H), 6.50 (d, *J* = 2.4 Hz, 1 H), 6.74 (d, *J* = 2.4 Hz, 1 H), 11.29 (s, 1 H). ^13^C-NMR (101 MHz, DMSO-D_6_): δ = 22.5, 38.5, 68.8, 69.5, 81.1, 100.0, 100.9, 102.4, 131.8, 139.2, 163.0, 165.8, 168.2.

### 3.9. Synthesis of (2R,3R,4aR)-2,3,4,4a-tetrahydro-2,3,7-trihydroxy-9-methoxy-4a-methylbenzo[c]chromen-6-one ((−)-altenuene, **1a**)

The native mycotoxin **1a** was synthesized analogously with 374 mg (1.16 mmol, 1.00 equiv) allylic alcohol **5b** and 580 mg (1.74 mmol, 1.50 equiv) boronate **4a**, yielding 222 mg (81%) (−)-altenuene (**1a**) as a white solid. R_f_ (DCM/MeOH, 20:1) = 0.18. MP = 117–120 °C. LC-MS (neg): [M − H]^−^ = 291, t_R_ = 16.67 min. HRMS (ESI): Calc. [M + Na]^+^ = 315.0839, found: 315.0836. ^1^H-NMR (400 MHz, DMSO-D6): δ (ppm) = 1.47 (s, 3 H), 1.95 (dd, *J* = 7.5, 14.1 Hz, 1 H), 2.26 (dd, *J* = 3.5, 14.0 Hz, 1 H), 3.70 (dd, *J* = 3.8, 7.6 Hz, 1 H), 3.86 (s, 3 H), 3.95 (dt, *J* = 4.4, 6.2 Hz, 1 H), 5.13 (d, *J* = 3.8 Hz, 1 H), 5.29 (d, *J* = 6.1 Hz, 1 H), 6.30 (d, *J* = 3.3 Hz, 1 H), 6.50 (d, *J* = 2.3 Hz, 1 H), 6.75 (d, *J* = 2.4 Hz, 1 H), 11.29 (s, 1 H). ^13^C-NMR (101 MHz, DMSO-D*_6_*): δ (ppm) = 27.4, 38.6, 55.9, 68.8, 69.5, 81.1, 100.0, 100.9, 102.3, 131.0, 131.8, 139.2, 163.0, 165.8, 168.2.

## 4. Conclusions

In summary, the successful optimization of a reported ALT (**1a**) synthesis provides efficient access to a novel deuterated derivative **1b**. Starting from commercially available D-*(−)*-quinic acid (**8**) as an inexpensive chiral pool compound, the D_3_-labeled natural product **1b** was obtained with an overall yield of 17% after nine steps. The newly synthesized ALT-D_3_ (**1b**) proved to be suitable as an internal standard for SIDA LC-MS/MS and thus ensures the availability of appropriate methods for the reliable screening of foods and feeds. Moreover, the optimized procedure may facilitate the development of new anti-tumor drug candidates with (−)-altenuene (**1a**) being a possible lead compound.

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
