# Peer review of "First Synthesis of (−)-Altenuene-D3 Suitable as Internal Standard for Isotope Dilution Mass Spectrometry"

_molecules, 2019, doi:10.3390/molecules24244563_

Round 1

Reviewer 1 Report

This manuscript describes the synthesis of deuterium labelled altenuene, known as a mycotoxin, as an internal standard for analyze the toxin by LC/MS method. The synthesis was achieved in 9 steps from easily available D-quinic acid. I would like to support this manuscript as a full paper for Molecule after the minor revision. Comments are as follows.

15,33,48  ---- Altenuen---  =>  ---altenuene---

60 MITSUNOBU    PODLECH =>  Mitsunobu Podlech

67 GRIGNARD => Grignard

92 Retention time data of 5a, 5b, and their epimer for GCMS are required in Table and experimental section.

92 Why are the product yield of 5b under the optimized conditions moderate to low ?

97 The authors attempted various conditions for the Suzuki coupling reaction. The details should be reported in the main text.

105 H2O =>  <subscript>2

125 Retention time of 1a & 1b are required in figure 2  (or main text) and experimental section.

134-135 Alternariol, Tentoxin --- => alternariol, tentoxin ---

Spelling out is needed for the following abbreviation CH, EE 

13C NMR signals (ppm and coupling constant) of the tri-deuterated products are required.

J   => <ithalic>J

The authors are advised to check the abbreviation format of the following journals.

Tetrahedron Letters, Mycotoxin research, PNAS, Mycotoxin Journal, Analytical and Bioanalytical Chemistry

Author Response

Dear Reviewer,

Thank you very much for your kind remarks on our article on the synthesis of (−)-altenuene-D3 as internal standard for SIDA.

I will do all the required format changes, add the missing information and will try to sufficiently reply to your comments in the following.

Point 1: Why are the product yield of 5b under the optimized conditions moderate to low?

Reply to Point 1: The best diastereomeric ratio was achieved at -78 °C (table 1, entry 8) but came with uncomplete conversion. Since the Rf values of the starting material and the desired alcohol 5b are quite similar during the separation conditions we raised the reaction temperature shortly after the addition of MeLi and went for a complete conversion at the cost of the diastereomeric ratio. Thus led to a diastereomeric ratio product of ~1.1:1 within the crude product in favor of 5b plus traces of the starting material 10b. The separation of the diastereomers furnished the desired compound 5b and pure 5a as well as a notable mixed fraction of both compounds which was not further purified. With purification of this mixed fraction the total isolated yield may have reached around ~45% but we didn’t pursue it because of a lack of time.

Point 5: The authors attempted various conditions for the Suzuki coupling reaction. The details should be reported in the main text.

Reply to Point 5: We did not test a lot of different conditions for the Suzuki coupling. After the conditions reported by Podlech et al. did not work we did a detailed literature research for the coupling of aryl boronates with halogenated allylic alcohols. The work of Lan et al. was the first hit and the given conditions worked quite well.

Point 6: 13C NMR signals (ppm and coupling constant) of the tri-deuterated products are required.

Reply to Point 6: The Signals of the deuterated carbons atoms could not be observed in the 13C-NMR spectra. We increased the sample amount within the limits of their availability, but we were still not able to detect the respective carbon signals and thus were not able to state them in the text. We also synthesized the native compounds whose NMR date was in total accordance with the deuterated compounds except for the deuterated 13C-signals (please see 13C-NMR spectra of 1a and 1b in the supporting materials). In the end the HR-MS measurements indicated the presence of the CD3-groups which was sufficient for us.

Yours sincerely,
Michael Sebald

Reviewer 2 Report

The manuscript by Sebald M.A. et al. reported a total synthesis of (−)-Altenuene and its stable isotope labeled derivative suitable for the implementation in a LC-MS/MS method for mycotoxin analysis. The work is well written and consistent. The scope is clear and the synthetic effort is acknowledged. Some minor comments are reported for the improvement of the manuscript.

The description of Table 1 should be corrected.

At line 85 the description of bromo derivative 10b seems to be referred to the entry 8 and not 11.

Correct or clarify the sentence.

Again regarding the Table 1, in my opinion that it needs to be discussed more in details. For example, the best stereoselective condition should be emphasized in the text.

The sentence at lines 94-96 is not refered to the scheme 3.

The conclusions are very short.

Author Response

Dear Reviewer,

Thank you very much for your remarks on our article on the synthesis and analytical evaluation of (−)-altenuene-D3 as internal standard for SIDA.

I will all the required format changed and extend the description of table 1 as well as the conclusion to emphasize the findings of our work.

Yours sincerely,
Michael Sebald

Reviewer 3 Report

This manuscript describes the synthesis of a deuterated derivative of altenuene as an internal standard for the suitable isotope dilution assay (SIDA).  Although some improvements in the reaction conditions, the synthetic scheme is totally identical with that reported by Altemoller and co-workers as Ref. 8.   

I could not find any novelty in this manuscript, and do not recommend for publication in Molecules.

Author Response

Dear Reviewer,

Thank you very much for your remarks on our article on the synthesis and analytical evaluation of (−)-altenuene-D3 as internal standard for SIDA.

Point 1: This manuscript describes the synthesis of a deuterated derivative of altenuene as an internal standard for the suitable isotope dilution assay (SIDA). Although some improvements in the reaction conditions, the synthetic scheme is totally identical with that reported by Altemoller and co-workers as Ref. 8.

Reply to Point 1: The significance of our work does not consist in having optimized a known synthetical approach towards the natural product but to find an efficient access to a deuterium labeled derivative and to proof it’s suitability as internal standard for SIDA. In addition we were able to optimize the synthesis of Podlech et al. in regard to the key intermediate alcohol (38% vs. 18%), the crucial Suzuki coupling protocol (81% vs. 70%) as well as the deprotection step at the end (81% vs. 55%) leading to a significantly increased overall yield of 17% after 9 steps (compared to 3% after 8/9 steps)

Point 2: Extensive editing of English language and style required

Reply to Point 2: We were surprised to hear that our article requires extensive editing of English language and style as we had a native speaker check the text. We will present the text to another native speaker to make sure it meets the high standards of written scientific communications.

Yours sincerely,

Michael Sebald

Round 2

Reviewer 3 Report

I accept the author's claims and recommend this manuscript for publication in Molecules.